# Comparison of Effectiveness Using Different Dual Bronchodilator Agents in Chronic Obstructive Pulmonary Disease Treatment

**DOI:** 10.3390/jcm10122649

**Published:** 2021-06-16

**Authors:** Shih-Lung Cheng

**Affiliations:** 1Department of Internal Medicine, Far Eastern Memorial Hospital, New-Taipei City 22060, Taiwan; shihlungcheng@gmail.com; Tel.: +886-2-8966-7000 (ext. 2160); Fax: +886-2-7738-0708; 2Department of Chemical Engineering and Materials Science, Yuan-Ze University, Taoyuan City 320315, Taiwan

**Keywords:** chronic obstructive pulmonary disease (COPD), dual bronchodilator, LABA/LAMA

## Abstract

The effectiveness and safety of fixed dual long-acting bronchodilators for chronic obstructive pulmonary disease (COPD) patients have been well established; however, there is a paucity of clinical effectiveness comparison in patients with COPD treatment. The aim of the current study was to compare the effectiveness of three once-daily dual bronchodilator agents in patients with COPD. Patients with diagnosed COPD and treated with a long-acting beta-agonist (LABA) + long-acting muscarinic antagonist (LAMA) fixed-dose combination therapy (UME/VIL (umeclidinium and vilanterol inhalation powder), IND/GLY (indacaterol and glycopyrronium), and TIO/OLO (tiotropium and olodaterol)) were enrolled in this retrospective study over a period of 12 months. Effectiveness assessments were evaluated using a COPD assessment test (CAT) and lung function parameters. Besides, times for acute exacerbation were also assessed. The enrolled patients’ number was 177 in IND/GLY, 176 in UME/VIL and 183 in TIO/OLO. Lung function measurements with FEV1 had significantly improved for patients using TIO/OLO (98.7 mL) compared to those of IND/GLY (65.2 mL) and UME/VIL (64.4 mL) (*p* < 0.001). CAT scores were also significantly decreased in patients treated with TIO/OLO (CAT down 5.6) than those with IND/GLY (3.8) and UME/VIL (3.9) (*p* = 0.03). Acute exacerbation was also reduced in patients using TIO/OLO (4.9%) compared with those using IND/GLY (10.2%) and UME/VIL (11.9%) (*p* = 0.01). Significant improvement in pulmonary function, symptoms were demonstrated after 12 months of LABA/LAMA fixed-dose combination therapy with three different treatment options. TIO/OLO demonstrated higher therapeutic effects compared with UME/VIL or IND/GLY. Determining clinical relevance will require a well-designed randomized controlled trial.

## 1. Introduction

Chronic obstructive pulmonary disease (COPD) is a primary driver of morbidity and mortality globally, with estimates placing it as a leading cause of death by 2020 [1,2]. The Asia-Pacific region reports the highest burden of COPD in terms of mortality, years spent living with disability, and years of life lost [3]. In Asia, the prevalence of COPD is high (6.2% in 2012), with a similar estimated prevalence in Taiwan (approximately 6.1–9.5%) [4,5]. Mortality due to COPD has risen gradually over time in Taiwan [6].

Pharmacologic therapy aims to reduce symptoms of COPD, increase exercise capacity, and improve patient quality of life and prognosis [7]. Bronchodilator agents are the cornerstones for COPD treatment. Long-acting beta-agonist (LABA) and long-acting muscarinic antagonist (LAMA) bronchodilator agent provides an effective treatment option in the management of COPD and is recommended as the first-line fixed-dose combination (FDC) treatment option in the majority of symptomatic patients with COPD, either alone or in combination. Among pharmacological therapy, dual bronchodilator agents with a combination of LABA and LAMA was an important therapy strategy in advanced COPD status, especially in the FLAME study [8]. Patients with more symptoms and previously frequent exacerbations (Group D) should be treated with dual bronchodilator agents initially, according to the 2017 revised GOLD guidelines [9]. Besides, other dual bronchodilator agents, such as UME/VIL (umeclidinium and vilanterol) [10] and TIO/OLO (tiotoprium and olodaterol) [11], also have the efficacy for reducing exacerbation rates and improved lung function status. However, evidence of the effectiveness of LABA/LAMA FDCs is limited, owing to scarce head-to-head comparison data among the Taiwanese population. Of the medications used for this study, UME/VIL, IND/GLY (Indacaterol + Glycopyrronium), and TIL/OLO are all fixed dual long-acting bronchodilators from two classes (LABA, long-acting β2 agonists + LAMA, long-acting muscarinic antagonists). The effectiveness of these medications on COPD patients in Taiwan remains unknown, and the aim of the study would compare the treatment effectiveness with different LABA/LAMA agents in clinical real-world practice.

## 2. Methods

### 2.1. Study Objectives

The study was a retrospective survey of the effectiveness among three dual bronchodilator agents. A total of 536 patients with diagnosed COPD and treated with a single inhaler combining a LAMA and a LABA. These dual bronchodilator agents included UME/VIL (umeclidinium and vilanterol inhalation powder), IND/GLY (indacaterol and glycopyrronium), and TIO/OLO (tiotropium and olodaterol). All patients were from Far Eastern Memorial Hospital (FEMH) in Taiwan and were examined and treated over a period of 12 months. All patients were over 40 years of age, current or former smokers (smoking pack/day–year > 10), symptomatic (e.g., chronic airway symptoms and signs, such as coughing, breathlessness, wheezing, and chronic airway obstruction), and their pulmonary function was consistent with a diagnosis of COPD (FEV1/FVC < 70%, post-bronchodilator FEV1 < 80% predicted value, and GOLD II–GOLD IV). Patients were excluded from the current study for suspected or combined sleep apnea, presenting with a comorbid pulmonary disease or other uncontrolled systemic diseases, abnormal chest X-ray results, evidence of alcohol abuse, or presenting with a lower respiratory tract infection treated with antibiotics and systemic steroids. Patients did not consent to participate in the study due to medical record review and Institutional Review Board waiver of consent. Patients were excluded if poor adherence and poor skills of inhaled agents were recorded from medical records.

### 2.2. Evaluation of Therapeutic Effects

All medical records were reviewed and analyzed. Patients were examined for the frequency of acute episode of symptomatic COPD, including a COPD assessment test (CAT) and evaluation of lung function (forced expiratory volume in 1 s (FEV1) and forced vital capacity (FVC)).

All parameters were compared at the time of the 12th month of treatment minus those at the time of study enrollment. Lung-function-dependent variables were defined using the difference between baseline values and values obtained after 12 months of treatment. Differences in CAT scores were calculated by subtracting CAT at 12 months from baseline CAT. Moreover, times for acute exacerbation (AE) were also measured during the treatment period and compared the AE rates difference in these three drugs’ patients.

### 2.3. Statistical Analysis

Descriptive statistics for treatment medication, demographics, comorbidities and clinical outcomes are provided. Continuous variables are reported as mean and standard deviation (SD). Chi-square tests of independence and ANOVA F-tests were conducted for each variable and treatment medication. ANOVA multiple comparisons, using Tukey’s HSD (Honest Significant Difference) tests, were conducted between users of different medications for all clinical outcomes. All statistical analyses were performed using SAS^®^ version 9.4 (Windows NT version, SAS Institute, Inc., Cary, NC, USA) and R (version 3.4.2; R Foundation for Statistical Computing, Vienna, Austria). A two-tailed *p*-value less than 0.05 was considered significant.

## 3. Results

The total enrolled patients were 177 in IND/GLY, 176 in UME/VIL, and 183 in TIO/OLO, respectively. Most patients in the current study were male (88.1%), with an average age of 70.51 years, mean weight and height of 73.15 kg and 166.13 cm, respectively (Table 1). Patients reported smoking a mean of 37.2 packs per year; however, this varied between patients (SD = 22.08). Furthermore, nearly half of all patients had diabetes (46–53%), and 34–37% had hypertension. Coronary artery disease and chronic heart failure were present in 27–32% of the study population. Medication type was independent of the presence or type of comorbidity.

There were no significant differences observed in FEV1 and FVC changes at baseline (before the study) between patients treated with different medications. After the study, there were significant improvements in pulmonary function parameters and symptom relief in all three medications after 12 months of treatment. Among these dual bronchodilators’ treatment effectiveness, patients treated with TIO/OLO achieved greater lung function improvement compared with those with the other two dual bronchodilator agents. Patients treated with TIO/OLO had the highest FEV1 (98.7 ± 38 mL) and FVC (127.3 ± 39.4 mL) improvement compared with baseline status than those with IND/GLY (FEV1: 65.2 ± 23.8 mL; FVC: 58.2 ± 35.7 mL) and UME/VIL (FEV1: 64.4 ± 24.1 mL; FVC: 79.1 ± 35.2 mL) (*p* < 0.001). (Table 2). Significant differences in FEV1 and FVC improvements after 12 months of treatment were observed (Figure 1 and Figure 2). Patients treated with IND/GLY and UME/VIL did not differ significantly in terms of FEV1 and FVC improvement; however, TIO/OLO users did have significantly increased lung function parameters after 12 months of treatment (Figure 1 and Figure 2).

Additionally, quality of life with symptoms using CAT score improved in all three groups. The difference (pre-treatment vs. post-treatment) in patients was IND/GLY (14.8 ± 5.9 vs. 10.9 ± 6.1, *p* = 0.03), UME/VIL (15.2 ± 5.8 vs. 11.3 ± 5.4, *p* = 0.02), and TIO/OLO (15.7 ± 6.3 vs. 10.1 ± 5.1, *p* = 002) (Table 2). There was greater improvement in patients treated with TIO/OLO compared with IND/GLY or UME/VIL (Figure 3). CAT score decreases were more pronounced in patients with TIO/OLO treatments (−5.6 ± 2.6) than patients treated with IND/GLY (−3.8 ± 2.5) and UME/VIL (−3.9 ± 2.2) (*p* = 0.03) (Table 2). Therefore, the quality of life and symptoms relief were more improved in patients with TIO/OLO compared to those with the other two dual bronchodilator agents.

The frequency of acute exacerbation was also reduced, and significant differences in patients using TIO/OLO (4.9%) compared with those using IND/GLY (10.2%) and UMEC/VIL were observed (11.9%) (*p* = 0.01). There was still the treatment benefit for decreased AE frequency in patients with TIO/OLO treatment (Table 2).

## 4. Discussion

The current non-randomized study compared the effectiveness of three LABA/LAMA fixed-dose combination therapies in patients with COPD from Far Eastern Memorial Hospital in Taiwan over a period of 12 months. CAT score, FEV1, and FVC improved after treatment with UME/VIL, IND/GLY, and TIO/OLO. Results revealed that clinical outcomes after 12 months of treatment (CAT, FEV1, and FVC differences) were significantly affected by the type of COPD medication used. All lung function measurements and symptoms were significantly more improved for users of TIO/OLO compared to users of IND/GLY and UME/VIL. The differences between the effects of UME/VIL and IND/GLY were not as pronounced. Besides, there was also statistical significance for decreased exacerbation rate in patients with TIO/OLO treatment.

Previously published studies have demonstrated the effectiveness and safety of fixed dual long-acting bronchodilators for COPD patients, specifically TIO/OLO. TIO/OLO is indicated for the maintenance treatment of airflow obstruction in adults with COPD [12]. This indication is supported by results from multiple randomized phase III studies of varying duration (6–52 weeks) [11,13,14]. TIO/OLO maintenance therapy improved lung function to a greater extent than the individual components or placebo and provided clinically meaningful improvements in health-related quality of life and dyspnea in 12- and 52-week studies. Tiotropium/olodaterol consistently improved 24 h lung function in 6-week studies, providing greater benefits than the monotherapies, placebo or twice-daily fixed-dose fluticasone propionate/salmeterol. Worth noting, in the subgroup analysis from Tonado [14] and DYNAGITO study [15], combination therapy with TIO plus OLO was more effective than TIO in reducing exacerbations in the Japanese population compared with other races. Taken together, including our results, TIO plus OLO combination therapy may have superior treatment effectiveness in the Asian population.

Current therapeutic policies for COPD management focus on improvement of symptoms, reduction of the risk for acute exacerbations, and reduced prognosis or mortality. Based on the treatment benefits evidenced, dual LAMA/LABA combinations play an essential role in stable COPD therapy. Long-acting inhaled bronchodilators have been demonstrated for their potential to reduce COPD exacerbations in several studies [16,17,18,19]. Among these studies have shown that the LAMA tiotropium (TIO) has greater efficacy against exacerbations than LABAs (POET-COPD [16] INVIGORATE studies [17]). In addition, the efficacy of TIO in reducing exacerbations was shown to be non-inferior to that of fixed-dose combination therapy with inhaled corticosteroids (ICS) and LABA salmeterol (INSPIRE study) [20]. In the SPARK study, fixed-dose combination therapy with IND/GLY was not superior to TIO monotherapy in reducing moderate and severe exacerbations [21]. Moreover, there were no significant differences in symptoms, health status, or risk of exacerbation between UME plus VIL and TIO [22]. Therefore, TIO was not a superior treatment for symptom relief and reduced exacerbation compared with ICS/LABA or IND/GLY and UME/VIL.

Olodaterol (OLO) is a novel once-daily LABA bronchodilator that was effective in lung function improvement [23,24,25]. The combination of TIO and OLO provides additional advantages in lung function and improves health-related quality of life [26,27,28]. The DYNAGITO study was performed to compare the safety and efficacy of TIO/OLO dual therapy versus TIO monotherapy in reducing exacerbations in COPD patients with a history of at least one exacerbation in the previous 12 months, and TIO plus OLO combination therapy provided a numerically greater reduction in moderate-to-severe exacerbations than TIO monotherapy [15]. Besides, anti-inflammatory effects had a central role in reducing AE and, combined with TIO plus OLO, had the synergistic effect for reducing neutrophilic inflammation in an in vitro study [29]. In fact, ideal dual bronchodilator agents should have these benefits, including the decrease in hyperinflation and mechanical stress, the modulation of mucus production and mucociliary clearance, the improvement of symptoms fluctuation and severity, and some potential direct and indirect anti-inflammatory properties [30]. From bench research, clinical trials to our real-world study, we inferred that TIO alone is non-inferior to the listed dual-therapies; therefore, it is expected that adding a LABA on top of TIO would cause an actual improvement (superiority) compared to UME/VIL and IND/GLY.

Another treatment effectiveness reason is the inhaled device. Several inhaled devices available in the treatment of COPD patients using dual bronchodilator agents, including Breezhaler device for IND/GLY, Ellipta device for UME/VIL and Respimat device for TIO/OLO. Ciciliani and their colleagues had compared these devices combining in vitro mouth–throat deposition measurements, cascade impactor data, and computational fluid dynamics (CFD) simulations, and Respimat showed the lowest amount of particles depositing in the mouth-throat model and the highest amount reaching all regions of the simulation lung model [31]. Therefore, TIO/OLO via Respimat with greater pharmacodynamic efficacy and inhaled device benefits achieved the treatment superiority for COPD compared with the other two bronchodilator agents.

In the daily clinical practice at our medical center, the patients have actually checked the skills of an inhaler device and detailed education to make sure that the inhaled skills were correct for every patient inspective of using different devices (Breezhaler, Novartis, Basel, Switzerland; Ellipta, GlaxoSmithKline, Brenford, UK; Respimat, Boehringer, Ingelheim, Germany). We also made sure that each patient exhaled completely and measured the peak inspiratory flow (PIF) for the dry powder inhalers. We would exclude any patients due to poor technique and poor adherence. Therefore, the effectiveness of treatment could be unrelated to poor device skills or poor adherence.

Previous studies have also been head-to-head compared the effectiveness of different bronchodilators in COPD patients. Feldmen and colleagues had reported that superiority was observed for the primary endpoint of trough FEV1 at week 8 with UME/VIL compared with TIO/OLO in patients with symptomatic COPD [32,33]. There are some differences in our study compared with Feldmen’s study. First, baseline patients’ lung function was different. Our patients’ FEV1 was 45% predicted, while Feldmen’s study FEV1 was 59.6% predicted. Second, the period of study was different. Our period of study was 52 weeks, and Feldmen’s study was 8 weeks. Third, acute exacerbation rate was evaluated in our study, but no data is reported on this in Feldmen’s study. Another head-to-head study had shown that IND/GLY and UME/VIL provided clinically meaningful and comparable bronchodilation [34]. Compared with rescue and adherence medication, UMEC/VI was superior to TIO/OLO for rescue medication use, and UMEC/VI initiators had better medication adherence [35]. For cost-effectiveness, TIO/OLO showed a cost-effective option for the maintenance treatment of adults with chronic obstructive pulmonary disease in the UK than the other two dual bronchodilators [36]. However, several review studies have documented that there were no significant differences among the LAMA/LABA combinations in terms of the number of moderate to severe exacerbations, all-cause mortality, major adverse cardiovascular events, or pneumonia [37,38,39,40]. Additionally, we compared the effectiveness from clinical trials of these bronchodilators, and we found that fewer inhaler errors, shorter instruction time, and patient preference was the Ellipta device (UME/VIL) [41]. The large IGNITE clinical trial program has established the efficacy of IND/GLY across different outcomes in COPD patients of all severities [42]; the Tonado program with OLO/TIO has also provided a large amount of data about the efficacy and safety of this combination in COPD [43,44].

We will also try to explain the effectiveness differences. First, the device was different. Respimat was soft-mist inhaled, but the other two inhalers were dry powder inhalers. The lung deposition would be better with Respimat than those with dry powder inhalers [31]. Second, the components were different. TIO/OLO had the synergistic effect for suppression of neutrophilic inflammation [29], but few studies had confirmed the neutrophilic inflammation with IND/GLY and UME/VIL. Third, the difference in receptors binding. In a previous study, TIO had the long dissociation half-life in muscarinic receptor-3 receptor residence time than those in GLY and UME [45]. It would infer that TIO has more of an anticholinergic effect than the other two drugs. There were different results in different studies for three drugs comparison; however, we thought that TIO/OLO might have more effects on lung hyperinflation and mechanical stress, inflammation, excessive mucus production with impaired mucociliary clearance, and symptom severity than IND/GLY and UME/VIL. The effectiveness discrepancy for these reports may be due to patient selection, study design, treatment duration and baseline medication usage. The definitive effectiveness of different treatments can only be performed through direct comparison in head-to-head RCTs. In the absence of such data, this indirect comparison may be of value in real-world clinical practice. Whether this effectiveness is clinically relevant will require further well-designed and large sample sizes in randomized studies.

Several limitations were noted in this study. First, this is not a prospective randomized study to compare these treatments’ effectiveness. Some bias could be detected, such as data collection, baseline medications, including comorbidity, which could influence the outcomes rather than the treatment itself. Their managing clinicians may have had biases about the treatments that each patient received, which could have again influenced the results. Our study was collected over 10 chest physicians’ medical records, and there was no significant discrepancy between the enrolled patients in each group. Besides, the duration for a medical record was one year, and we could complete the collected data. It means that the treatment’s effectiveness may be not related to clinicians, inadequate data collection, or a short duration of study time in our study. Validating the current results in terms of their clinical relevance will require further well-designed randomized controlled trials. Second, there was a relatively small population to enroll in the study. It should include a large population to confirm these results. Third, the study period was only 12 months. It should take too long to prove these treatment’s effectiveness. Fourth, the study performance was only one center, and it may have potential bias. The study was the first head-to-head comparison of the effectiveness in these different kinds of LABA/LAMA fixed combination agents, though there were some limitations and biases.

## 5. Conclusions

This report is a brief summary of three fixed dual long-acting bronchodilators (UME/VIL, IND/GLY, and TIO/OLO) used on patients with COPD in a Taiwanese medical center (FEMH). Significant improvement in pulmonary function parameters and symptom relief was found in all three medications after 12 months of treatment. Most importantly, TIO/OLO demonstrated higher therapeutic effects compared with the two other drugs, especially in the reduction of acute exacerbation. Whether this effectiveness is clinically relevant will require further well-designed randomized studies to confirm the treatment benefits.

## Figures and Tables

**Figure 1 jcm-10-02649-f001:**
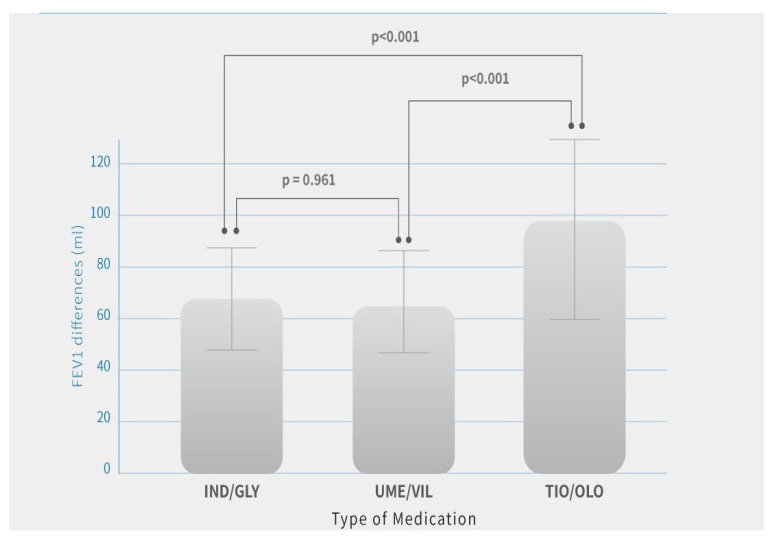
FEV1 differences after 12 months of treatment by COPD medication. Note: Means are shown in the bar chart; *p*-values were generated using Tukey’s HSD (Honest Significant Difference) for comparisons between users of different medication, *p* < 0.05 was considered significant (*p* < 0.001).

**Figure 2 jcm-10-02649-f002:**
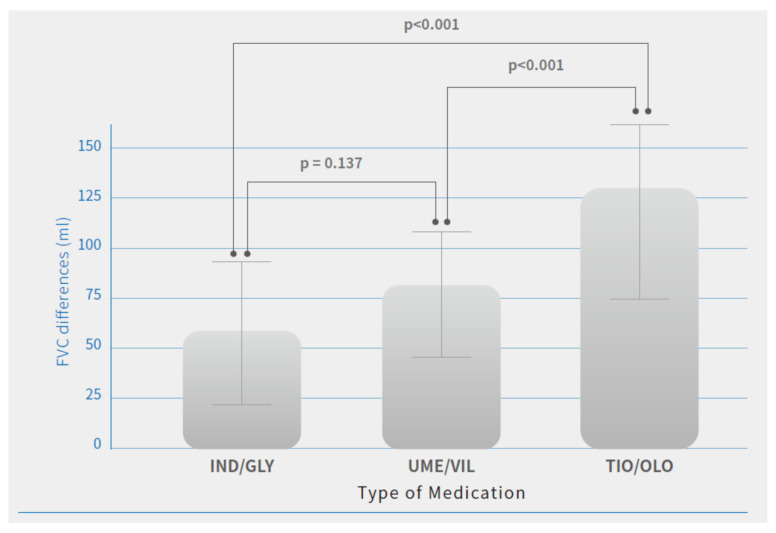
FVC differences after 12 months of treatment by COPD medications. Note: Means are shown in the bar chart; *p*-values were generated using Tukey’s HSD (Honest Significant Difference) for comparisons between users of different medication, *p* < 0.05 was considered significant (*p* < 0.001).

**Figure 3 jcm-10-02649-f003:**
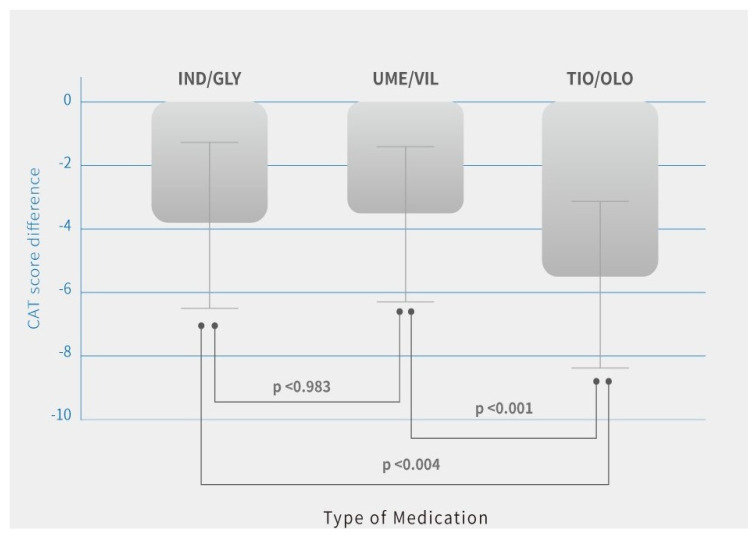
CAT score differences after 12 months of treatment by COPD medications. Note: Means are shown in the bar chart; *p*-values were generated using Tukey’s HSD (Honest Significant Difference) for comparisons between users of different medication, *p* < 0.05 was considered significant (*p* < 0.001, *p* < 0.004).

**Table 1 jcm-10-02649-t001:** Descriptive statistics of demographics, clinical conditions and medications in COPD patients.

Characteristics	COPD Medication			*p*-Value ^1^
	Ultibro (*n* = 177)	Anoro (*n* = 176)	Spiolto (*n* = 183)	
Demographics				
Sex				
Female	20 (11.3)	26 (14.8)	18 (9.8)	0.47
Male	157 (88.7)	150 (85.2)	165 (90.2)	0.61
Age	71.4 ± 7.37	71.3 ± 7.6	69.6 ± 7.8	0.47
Weight	72.3 ± 6.75	73.8 ± 7.6	73.3 ± 6.7	0.39
Height	167.5 ± 7.9	168.2 ± 5.01	165.2 ± 4.9	0.18
Smoking (pack/year)	36.9 ± 16.8	37.8 ± 27.94	36.6 ± 19.3	0.72
Comorbidities				
Diabetes	87 (49.2)	81 (46.0)	97 (53.0)	0.17
Hypertension	67 (37.8)	60 (34.1)	68 (37.1)	0.32
Coronary artery disease	58 (32.8)	51 (28.9)	53 (28.9)	0.75
Chronic heart failure	49 (27.7)	53 (30.1)	55 (30.1)	0.48

Note: Counts and proportion were presented for categorical variables and mean ± SD for continuous variables. ^1^
*p*-values from Chi-Square Tests of Independence for categorical variables and ANOVA f-test for continuous variables.

**Table 2 jcm-10-02649-t002:** Descriptive statistics of clinical outcomes by COPD medication.

Stratified by Type of Medication	COPD Medication			*p*-Value ^1^
	Ultibro (*n* = 177)	Anoro (*n* = 176)	Spiolto (*n* = 183)	
Lung function parameters				
FEV1 base (L)	1.41 ± 0.18	1.4 ± 0.13	1.37 ± 0.17	0.64
FEV1 base (% of predicted)	46.74 ± 11.9	45.72 ± 8.0	45.59 ± 9.3	0.81
FEV1 12 months (L)	1.42 ± 0.16	1.45 ± 0.14	1.47 ± 0.16	0.52
FEV1 difference after 12 months (mL) ^2^	65.2 ± 23.8	64.4 ± 24.1	98.7 ± 38.0	<0.0001
FVC base (L)	2.35 ± 0.26	2.36 ± 0.19	2.39 ± 0.22	0.61
FVC base (% of predicted)	47.6 ± 10.9	47.7 ± 7.2	48.1 ± 8.4	0.75
FVC 12 months (L)	2.42 ± 0.48	2.43 ± 0.19	2.52 ± 0.21	0.63
FVC difference (mL) ^2^	58.2 ± 35.7	79.1 ± 35.2	127.3 ± 39.4	<0.0001
Symptoms Scores				
CAT base	14.8 ± 5.9	15.2 ± 5.8	15.7 ± 6.3	0.74
CAT 12 months	10.9 ± 6.1	11.3 ± 5.4	10.1 ± 5.1	0.37
CAT score difference ^2^	−3.8 ± 2.5	−3.9 ± 2.2	−5.6 ± 2.6	0.03
Acute exacerbation of COPD				
No	159 (89.8)	155 (88.1)	174 (95.1)	0.58
Yes	18 (10.2)	21 (11.9)	9 (4.9)	0.01

Note: Counts and proportion were presented for categorical variables and mean ± SD for continuous variables. ^1^
*p*-values from Chi-Square Tests of Independence for categorical variables and ANOVA f-test for continuous variables. ^2^ FEV1, FVC, and CAT score differences were calculated by subtracting the FEV1, FVC and CAT score after 6 months of treatment to the base FEV1, FVC and CAT score.

## Data Availability

The data will not be shared with a reason.

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
