# Peer review of "Comparison of Effectiveness Using Different Dual Bronchodilator Agents in Chronic Obstructive Pulmonary Disease Treatment"

_jcm, 2021, doi:10.3390/jcm10122649_

Round 1

Reviewer 1 Report

Properly conducted double blinded randomized controlled studies are important to assess the efficacy of treatments, particularly medications, before widespread recommendations for their use can be made.  However, as these studies are tightly controlled and the subjects in them are carefully selected, they may have limitations in their applicability to patients who may have the disease in question (such as COPD), but would not have been eligible for inclusion in the efficacy studies; and they may have limitations in clinical settings, which do have the same control over various factors such as adherence and inhaler techniques as occurs in efficacy studies.

Once a treatment’s efficacy has been established, it is thus important to conduct effectiveness studies in a broader range of patients in clinical settings, which do not have the same degree of control as in efficacy studies. These effectiveness studies will help to tell clinicians if the treatment in question has broader applications in general clinical settings.

The study by Cheng is clearly an effectiveness study, rather than an efficacy study, and this should be made clearer in the manuscript, especially when it is stated that the aim is ”to compare the efficacy of three once daily dual bronchodilator agents in patients with COPD”.

This is a retrospective study, which, presumably means that the data was collected in hindsight rather than prospectively, so it is prone to significant biases. As the patients were not randomized to receive one of the three treatments, there have been biases in how patients were given a certain treatment, which could influence the outcomes rather than the treatment itself. As the patients were not blinded, both they and their managing clinicians may have had biases about the treatments that each patient received, which could have again influenced the results. These issues should be addressed in the Discussion.

In the Discussion, the author mentions some of the definitive efficacy studies for these three COPD treatments. It would be useful to make a qualitative comparison of the improvements achieved in this study by Cheng and the randomized controlled studies. This would then tell the readers whether or not the benefits obtained in a controlled trail setting could also be achieved in usual clinical practice.

Another factor is that the drugs are given by different inhalers, so it is not clear if any differences are due to the drugs themselves or the inhalers used for their delivery. This is noted by the author, but need more detailed discussion.

As is pointed out at the end of the Conclusion, prospective randomized controlled trails would be needed to really compare the three treatments. An advantage of Cheng’s study is that it provides data on possible differences in outcomes between the three treatments, which could guide power calculations for such studies. Would the author like to speculate in the Discussion about large controlled studies would have to be to compare these treatments?

Is there any data on adherence and/or inhaler technique?

Author Response

Dear Reviewer 1: 

Dear Reviewer, 1:

We are very grateful for your expert comments and valuable recommendation. The manuscript has been revised according to your suggestion. We will reply all suggestions and questions point by point for the manuscript.  

We deeply appreciated the expert comment and considerate recommendation from you. We would like to thank you for your thorough review and excellent suggestion. Thanks again for reviewers’ kindness and give us the opportunity to revise our manuscript. Hope to have good news for acceptance.

Best regards

Shih-Lung Cheng, MD. PhD

Department of Internal Medicine, Far Eastern Memorial Hospital, Taipei, Taiwan

Phone: 886-2-89667000 ext 2160      

Fax: 886-2-77380708        

Reviewer 2 Report

The current study by Dr. Cheng compares and contrasts the efficacy of combined LAMA/LABA inhalation therapy in COPD in a Taiwanese population. Three different combinations were compared: umeclidinium/vilanterol, tiotropium/olodaterol, and glycopyrrolate/indacaterol. Of these three combinations, the combination tiotropium/olodaterol was most efficacious in the studied population.

I have some major concerns with this study that need to be addressed:

1). The discussion lacks a proper discussion of the current results with previous publications. For example, Feldman et al (PMID 29094315) demonstrated that umeclidinium/vilanterol was superior to tiotropium/olodaterol in improving trough FEV1 at week 8 in patients with COPD. A comprehensive inclusion of previous studies should be included and discussed.

2). The author should include a possible explanation for the observed differences in efficacy between the different combinations as well as compared to previous studies.

3). The data is not clearly described.

  • In the conclusion the author mentions 'statistically significant improvement in pulmonary function parameters, symptoms relief and exacerbation reduction were found in all three medications after 12 months of treatment' (page 7, lines 213-215). This must also be described in the results sections or the Tables/Figures, where it is currently lacking.
  • In the Tables, the p-value is not explained: what is compared?
  • It is unclear what is meant with the sentences 'there were no significant differences observed in FEV1 and FVC changes from baseline between patients treated with different medications.' (Page 3, lines 110-111) or with the sentence 'significant differences in FEV1 and FVC improvements after 12 months of treatment were observed.' (Page 3, lines 116-117).
  • It seems that 'Figure 1' in line 117 should be be 'Figure 1 and 2'. The same is true for 'Figure 2' in line 120.
  • The mean pack-years of 36.38 (page 3, line 104) seems to be too low as the average in each study group is higher than that (Table 1).
  • There seems to be some words missing in the sentence 'CAT score decreases were significantly in patients with TIO/OLO treatments than patients treated with IND/GLY and UME/VIL' Page 5, lines 125-127). Do all treatments lower the CAT score compared to baseline and is this only significant for TIO/OLO or do all treatments lower the CAT score, but TIO/OLO has a stronger effect on this?

4). The data in Figures 1 and 2 should include the SD or SEM. It also seems that only the number in the Utibro bars (Figures 1 and 2) is presented in bold. Figure legends should include the explanation of the drug names.

5). It is unclear where the statement that 'Tio has treatment superiority for symptoms relief and reduced exacerbation compared with ICS/LABA or IND/GLY and UME/VIL' (page 6, lines 172-174) is based on. Does the author suggest that monotherapy with tiotropium is superior to the other listed combination therapies? The author also mentions that the 'differences between UME/VIL and IND/GLY were not as pronounced' (page 6, lines 143-144). For most parameters studies, there were actually no differences at all.

6). References are missing for the statement on page 7, lines 188-190 (starting with 'based on previous studies'). The statement on page 7, lines 198-200 should be compared and contrasted with other studies, as the combination of tiotropium/olodaterol has not shown superiority in all studies.

7). The text should be double checked for grammar and spelling.

Minor:

1). The author should state in the methods section that patients were treated with a single inhaler combining a LAMA and LABA.

2). The description of evaluation of therapeutic effects (page 2) seems to be incorrect. It seems to be that the data is presented as "12 months minus baseline" instead of "baseline minus 12 months"

3). The data is presented in the Tables/Figures as "Utibro, Anoro, Spiolto", but it is not explained what this represents. Instead, the data is described in the text using IND/GLY, UME/VIL, and TIO/OLO. This should be synchronized and explained. Also, the baseline FEV1 for Anoro (1.4) should have same number of decimals as the other 2 (1.41 and 1.37)

4). Be consistent with the abbreviations. For example, UMEC vs UME.

5). Make sure to double check the requirements of the journal:

  • End the introduction with a summary of the main findings
  • Put the references before the periods: [1]. instead of .[1]
  • A list of abbreviations is not needed.
  • A DOI should be included for each reference.

Author Response

Dear  Reviewer 2:

We are very grateful for your expert comments and valuable recommendation. The manuscript has been revised according to your suggestion. We will reply all suggestions and questions point by point for the manuscript.  

We deeply appreciated the expert comment and considerate recommendation from you. We would like to thank you for your thorough review and excellent suggestion. Thanks again for reviewers’ kindness and give us the opportunity to revise our manuscript. Hope to have good news for acceptance.

Best regards

Shih-Lung Cheng, MD. PhD

Department of Internal Medicine, Far Eastern Memorial Hospital, Taipei, Taiwan

Phone: 886-2-89667000 ext 2160      

Fax: 886-2-77380708        

Round 2

Reviewer 1 Report

no further comments

Author Response

Thanks for reviewer's comments.

Best regards

Shih-Lung Cheng

Reviewer 2 Report

See attached.

Author Response

(The authors gave the same response as above.)
